# A color image contrast enhancement method based on improved PSO

**Xiaowen Zhang, Yongfeng Ren** *, **Guoyong Zhen, Yanhu Shan, Chengqun Chu**

School of Instrument and Electronics, North University of China, Taiyuan, Shanxi Province, People's Republic of China

* 503212590@qq.com

**Data Availability Statement:** All relevant data are within the manuscript and its Supporting information files.

**Funding:** This work was supported by the National Nature Science Foundation of China (6177012376).

## Abstract

Image contrast enhancement uses the object intensity transformation function to maximize the amount of information to enhance an image. In this paper, the image enhancement problem is regarded as an optimization problem, and the particle swarm algorithm is used to obtain the optimal solution. First, an improved particle swarm optimization algorithm is proposed. In this algorithm, individual optimization, local optimization, and global optimization are used to adjust the particle's flight direction. In local optimization, the topology is used to induce comparison and communication between particles. The sparse penalty term in speed update formula is added to adjust the sparsity of the algorithm and the size of the solution space. Second, the three channels of the color images R, G, and B are represented by a quaternion matrix, and an improved particle swarm algorithm is used to optimize the transformation parameters. Finally, contrast and brightness elements are added to the fitness function. The fitness function is used to guide the particle swarm optimization algorithm to optimize the parameters in the transformation function. This paper verifies via two experiments. First, improved particle swarm algorithm is simulated and tested. By comparing the average values of the four algorithms under the three types of 6 test functions, the average value is increased by at least 15 times in the single-peak 2 test functions: in the multi-peak and multi-peak fixed-dimension 4 test functions, this paper can always search for the global optimal solution, and the average value is either the same or at least 1.3 times higher. Second, the proposed algorithm is compared with other evolutionary algorithms to optimize contrast enhancement, select images in two different data sets, and calculate various evaluation indicators of different algorithms under different images. The optimal value is the algorithm in this paper, and the performance indicators are at least a 5% increase and a minimum 15% increase in algorithm running time. Final results show that the effects the proposed algorithm have obvious advantages in both subjective and qualitative aspects.

## 1 Introduction

The contrast of an image is an important factor affecting its quality. The enhancement of the contrast of an image is a process that increases the dynamic range for visual effects and highlighting details. As image processing research deepens in the fields of vision, remote

**Competing interests:** The authors have declared that no competing interests exist.

sensing [1], biomedical image analysis [2], fault detection, and other fields, strong requirements for high-brightness, high-contrast, and high-detail digital images to produce visual natural images or to transform images, such as enhancing internal visual information, have become basic requirements for nearly all image processing tasks [3]. However, optical systems, cameras, and image capture systems under the effects of lighting cause low contrast in captured images, which cannot meet the requirements of engineering. Therefore, enhancing the contract of the image is particularly important. Contrast enhancement techniques can improve the perception of information in images or provide meaningful information for real-time image processing applications.

Owing to lighting or some other conditions (e.g., imaging device limitations or inappropriate exposure parameter settings), the captured images tend to suffer from low image contrast, and blurring, to name a few. These issues affect the collection of images in photography, forensics, analysis, surveillance, and some other optical imaging systems. Despite the astonishing advancements in image capture devices, various natural and artificial artifacts that result in poor quality of the captured images persist. Therefore, quality improvement over the original captured images is an essential part of image preprocessing. Low-light image enhancement technology aims to enhance the visual effect of images by highlighting blurred or even hidden details in the image, increasing the brightness and contrast of the image. This problem is relatively challenging and has become a hot research topic and has been extensively studied by scholars in recent years.

Given the importance of image contrast enhancement techniques, researchers have developed many algorithms. Currently, contrast enhancement methods can be divided into spatial and transform domain enhancements. Contrast enhancement in the spatial domain is based on grayscale transformations of non-linear functions, such as logarithmic transformation [4], gamma function [5], histogram-based technology [6], and non-linear secondary filtering [7]. Methods for enhancing the contrast in the transform domain include filtering methods [8]. Histogram equalization is a widely used spatial image enhancement method. An adaptive histogram equalization method is proposed in [9]. This method estimates the probability density of the original input image, and then the size of the function is scaled and the scale factor adjusted adaptively according to the average intensity value of the image. In [10], the researcher proposes a technology on the basis of joint histogram equalization, which uses information between each pixel and its neighboring pixels to improve the contrast of the image. The study posits a non-parametric modified histogram equalization that effectively handles histogram peaks, reduces distortion in smooth regions, and requires no empirical adjustment of parameters [11, 12]. The intensity value of the image can also be adjusted according to different contrast and sharpness measurements [13]. The method based on histogram equalization reduces the gray level of the image after the transformation, and certain details disappear. Some images, such as the histogram, have peaks, and the contrast is unnaturally enhanced after processing.

In most applications, automatic contrast enhancement technology is required without manual intervention. However, the automation of the algorithm is not a simple task, because it requires the evaluation of an objective function that measures the quality of the enhanced image. To this end, an optimization algorithm based on evolutionary calculations [13–15] was proposed to automate the contrast enhancement task. These techniques are used to determine the best parameter settings or the best input/output mapping to obtain the best quality images. One disadvantage of the image contrast enhancement algorithm based on evolutionary algorithms is that the algorithm iterative time is long, and the image processing time is slow. Aiming at this problem, this paper proposes a color image contrast enhancement algorithm on the basis of the improved particle swarm algorithm based on the concept of sparse penalty in

machine learning. While enhancing the image contrast, the method shortens the time required for the evolutionary algorithm to optimize the image. It also largely improves the image quality. This study provides a new idea for the enhancement of low-contrast images in the future.

The rest of this article is organized as follows. Section 2 sorts out the relevant literature, makes a theoretical comparison with the existing methods, and introduces the traditional PSO algorithm. Section 3 proposes an improved version of PSO and enhances image contrast on the basis of the improved PSO algorithm. Section 4 evaluates the method in extensive experiments. Finally, Section 5 concludes the paper.

## 2 Background

### 2.1 Related work

Contrast enhancement algorithms aim to provide an image with additional vivid colors and higher detail clarity. Contrast enhancement algorithms are closely related to different visual properties. Attributes include brightness, hue, color, hue, and saturation. Existing image enhancement techniques are mostly empirical or heuristic methods that are strongly correlated to specific types of images and usually aim to improve the contrast of degraded images during acquisition. In fact, finding the optimal grayscale map that adaptively enhances each different input image can be viewed as an optimization problem.

In recent years, many bionic algorithms for image enhancement have been developed, such as particle swarm algorithm(PSO), artificial bee colony algorithm, genetic algorithm, and cuckoo algorithm, among others.

Genetic algorithm [14] can be used as a preprocessing step to enhance the histogram distribution near the bimodal image and improve the effect of downstream image processing technology. Munteanu and Rose [15] proposed another genetic algorithm-based contrast enhancement method. This paper uses a four-parameter local/global transformation function and optimizes these parameters by genetic algorithm. The objective function combines the entropy of the enhanced image, the number of edge pixels, and the sum of edge intensity.

In [16], a new enhanced cuckoo search (ECS) algorithm is proposed to realize the automatic enhancement of image contrast by optimizing the local/ global transformation parameters. In the ECS image enhancement algorithm, the sum of the entropy, number of edge pixels, and edge strength are used to construct the objective function. A new local/ global enhancement transform parameter search range is proposed to overcome the problems of local/ global enhancement transforms in enhancing edges, and distorted images achieve better enhancement effects. A method for contrast enhancement of fingerprint images based on cuckoo search (CS) was proposed in [17]. In [17], the fingerprint image is enhanced using grayscale mapping transformation, and the grayscale between the input image and the enhanced image is mapped using the CS algorithm. Here, a parameter Pa called abandonment probability is used to update the solution in each iteration. The improved CS algorithm uses Cauchy mutation to reconstruct Pa worst nests. In [18], an adaptive image enhancement algorithm with hybrid CS and PSO (CSPSO) search algorithm was proposed. Here, an incomplete beta function is used as the boosting transform, and the worst CS set is updated using the PSO algorithm instead of reconstructing the worst set Pa such as the CS algorithm. Recently, an adaptive CS algorithm was proposed for satellite image contrast enhancement in [19], which has the same local/global enhancement transformation and parameter range as [14], with discovery and mutation randomization instead of fixed Pa, generates a new solution and proposes a medical image enhancement algorithm based on wavelet masking. In this paper, Pa is not fixed but adaptive through adaptive crossover and mutation process.

In [20], a new target fitness function is proposed for the indispensable fitness function in the enhanced image quality evaluation, including four performance indicators, namely, the sum of edge strengths, the edges of the number of pixels, image entropy, and image contrast. The fitness function automatically measures the quality of the generated image, and the artificial bee searches for the optimal conversion function under the guidance of the new cost function. Second, this paper uses the incomplete beta function as the image transformation function to guide the search action of artificial bees. Image enhancement based on firefly algorithm is proposed in [21] using the same transformation and objective function as [14]. A method for contrast enhancement of grayscale and color images using artificial bee colony algorithm is proposed. Furthermore, in [1], using the same local/global transformation functions and parameter ranges as [14], an improved differential evolution algorithm is proposed for contrast enhancement of satellite images. However, it uses a different fitness function that combines entropy, edge information, and standard deviation.

In [22], the intensity transform function serves as a traditional particle swarm algorithm to maximize the information amount of the enhanced image. A parametric transformation function that utilizes local and global information from the image is used. An objective criterion for image enhancement considering image entropy and edge information is proposed to achieve the best enhancement effect. Zhang [23] proposed a sonar image enhancement method on the basis of particle swarm optimization. Sonar images have important characteristics such as low resolution, strong echo interference, small target area, and blurred target edges. In this case, obtaining satisfactory results using the global image enhancement algorithm is difficult. In this paper, an adaptive local enhancement algorithm is proposed, which uses edge count, edge strength, and entropy to evaluate enhanced images. Particle swarm optimization (PSO) is used to determine the best enhancement parameters. Kanmani [24] proposed an optimized color contrast enhancement algorithm to improve the visual perception of information in images using an adaptive gamma correction factor selected by particle swarm optimization (PSO) to improve entropy and enhance image detail. And Kanmani [25] proposed a new Eigen face recognition method that uses particle swarm optimization (PSO), self-tuning particle swarm optimization (STPSO), and brainstorm optimization (BSO) to determine the optimal fusion coefficient. To fuse CT and MRI images. And they used PSO to optimize the weights to obtain the weighted average fusion image information, and the objective function jointly maximized the entropy and minimized the root mean square error to improve the image quality [26].

## 2.2 Particle swarm optimization algorithm

This section provides a brief introduction of the PSO algorithm. PSO algorithm [27] is an evolutionary algorithm. Its optimization process is performed by simulating the foraging phenomenon of bird flocks in nature. When bird flocks search for food within a certain range to find the optimal food source, the memory performance must remember the best food it passed and its location after communicating with its peers. After searching through the layers of birds, the birds are guided toward the optimal food location to find the most excellent food source.

The PSO algorithm optimization process is described briefly as follows:

**Initialize**: initialize particle swarm parameters, velocity range, space dimension, maximum iterations, population size, acceleration factor, etc.

**Update**: In the update iteration process, the speed and position update formula of traditional PSO is as follows:

$$v_{i,j}(t+1) = v_{i,j}(t) + c_1 r_{1,j}(t)\left[p_{b,j}(t) - x_{i,j}(t)\right] + c_2 r_{2,j}(t)\left[p_{g,j}(t) - x_{i,j}(t)\right], \qquad (1)$$

$$x_{i,j}(t+1) = x_{i,j}(t) + v_{i,j}(t+1), \tag{2}$$

where $x$ represents the position, $v$ represents the speed, and $p$ represents the optimal solution. The individual optimal solution is $p_{b,j}$, and the global optimal solution is $p_{g,j}$. $i$ represents the current particle, $j$ refers to the current dimension, $t$ is the current number of iterations, and $c_1$ and $c_2$ represent the acceleration factor of the learning factor. The value is usually between 0–2, and $r_{1,j} \sim U(0,1)$ and $r_{2,j} \sim U(0,1)$ represent two independent random functions.

**Determine**: if the iteration termination condition is met, terminate the iteration and output the global optimum. Otherwise, return to the previous step to continue the iterative calculation.

## 3 Proposed PSO-based image contrast enhancement approach

In summary, this paper uses the advantages of particle swarm algorithm with few parameters, fast search speed, and simple structure. It also proposes an image contrast enhancement method on the basis of improved particle swarm algorithm. Color image processing usually processes R-, G-, and B sub-channels but ignores the relevant information between the three sub-channels. In this paper, the color image is represented by a quaternion matrix, and the quaternion matrix is processed directly.

Second, the method uses an improved particle swarm algorithm to optimize the parameters and determine the optimal pixel value in the entire image intensity space. The traditional PSO-based image contrast enhancement method involves finding the optimal value in the solution space by using the global optimal and self-optimized adjusting particles to allow its easy fall into the position of the local optimal solution. This paper adds a topology structure to adjust the local optimal solution. The global optimal, local optimal, and individual optimal jointly adjust the particle's flying speed to avoid premature convergence. Intuitively, if the resolution of the image increases, the solution space of the particle swarm will increase. Therefore, the time complexity of the algorithm will increase considerably. To overcome this challenge, this paper adds a sparse penalty term to the particle swarm algorithm, thereby adjusting the sparsity of the algorithm and the size of the solution space.

Finally, a new fitness function is proposed. The chroma of a color image is the key to enhancing image contrast [20]. However, traditional PSO-based image contrast enhancement methods ignore this information. Hence, a new performance indicator is incorporated into the cost function of the method. The proposed new fitness function includes five performance indicators: (a) the sum of edge strengths, (b) the number of edge pixels, (c) the color entropy, (d) the image contrast, and (e) the chroma of the image. This fitness function automatically measures the quality of the generated image and determines the quality of the enhanced image. Therefore, the particle swarm optimization algorithm seeks the optimal transfer function under the guidance of the new cost function.

### 3.1 Proposed PSO

Through the description of the existing PSO algorithm, although a large number of research results have been obtained, problems persist such as being trapped easily in local optimum, premature convergence, and insensitivity to finding the optimal solution for multi-peak functions. For such issues, in this paper, we use the topology structure to improve the PSO and increase the diversity of particles. Topology [28] is typically used in optimization

algorithms to refer to the communication methods and neighborhood relationships among individuals in a group. It can control information transmission between particles. The most important thing in the PSO algorithm is the cooperative behavior of particles between groups. Through cooperation and communication between particles, their states change as they search for the global optimum. Topology can change the neighbors of particles and change communication between particles. The local optimal solution after topology communication is used as an item, and the particle's flight direction is adjusted together with the global and individual optimum.

Fig 1 shows that after the topology is initialized, the population is divided into several adjacent regions according to the subscripts of the particles, that is, particles $x_1$ and $x_2$ can be regarded as adjacent in an adjacent region with a radius of 1. Regardless of the spatial position of the particle, according to the spatial position of the particle, in each iteration of the algorithm, the distance between one particle and other particles in the population is calculated, and the variable $d_{max}$ is used to mark the maximum distance between any two particles; for each particle, the ratio of $\|x_a - x_b\|/d_{max}$ is also calculated, where $\|x_a - x_b\|$ is the distance from the current particle a to another particle b, using the calculated smaller ratio as the basis for selecting adjacent particles. The ratio is used as the basis for selecting adjacent particles. According to the selection of four nearest neighbors as a group, they are connected in pairs; they also communicate with each other and calculate the fitness value of each particle. According to the fitness value function, the optimal value directly enters the upper layer as the parent node. The calculation method is the same as that of the previous layer on the layer of the parent node. If redundant particles exist, they are directly placed in the next layer for communication and comparison, and so on, until the global optimum is found.

The biggest feature of the particle swarm optimization algorithm is to use the iterative process to solve the optimal value, but it takes a very long time to complete the iterative process. Therefore, this paper adds a sparse penalty term to improve this problem. The sparse penalty term [29] is a concept in machine learning. After comparing the individual optimal, local optimal, and global optimal in the particle swarm, some of the most inferior solutions will be obtained, that is, redundant terms. Using the sparse penalty term to eliminate the last solution in the next iteration, these least solutions can be ignored, the sparsity of the algorithm can be addressed, and the size of the solution space determined.

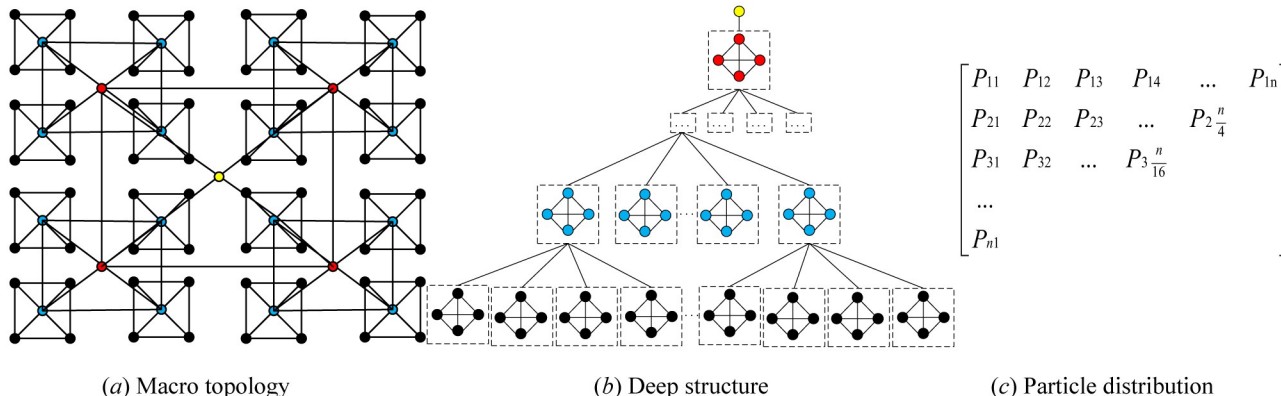

(*a*) Macro topology (*b*) Deep structure (*c*) Particle distribution

**Fig 1. Schematic diagram of local optimal iteration.** (*a*) Macro topology. (*b*) Deep structure. (*c*) Particle distribution.

Therefore, the global, local, individual optimal values and sparse penalty term are added to the speed and position update formula. The updated formula is as follows:

$$V_{i,j}(t+1) = k\left(\omega V_{i,j}(t) + c_1 r_1 \left(p_{b,j} - x_{i,j}(t)\right) + c_2 r_2 \left(p_{g,j} - x_{i,j}(t)\right) + c_3 r_3 \left(p_{l,j} - x_{i,j}(t)\right) + L_1\right), \quad (3)$$

$$x_{i,j}(t+1) = x_{i,j}(t) + v_{i,j}(t+1). \quad (4)$$

In Eq (3)), $k$ is the constriction factor, $\omega$ is the inertia weight, $p_{b,j}$ is the best solution of the individual particles, $p_{l,j}$ is the best solution of the particles in the local neighborhood, and $p_{g,j}$ is the best solution of the global particles that are recorded, where $L_1 = \lambda w_1$ are sparse penalty terms, $\lambda$ is the penalty coefficient, $w_1$ is the weight of the last solution, $c_1$ denotes adjusting the individual optimal acceleration coefficient, $c_2$ denotes adjusting the local optimal acceleration coefficient, $c_3$ denotes adjusting the global optimal acceleration coefficient, and $r_1, r_2, r_3 \in (0, 1)$ are independent of each other.

Notably, in the improved particle swarm algorithm, considering the different values obtained in each run, deviations are anticipated. Therefore, in the experiment, this paper tests each particle swarm algorithm in each test function. Averaged over 30 runs, when optimizing the four parameters in the transformation function, the average value obtained each time is used to optimize. Thus, in the data set experiment, only one result is obtained.

## 3.2 Proposed image contrast enhancement approach

For color image enhancement, this paper uses quaternions to represent color images. In subsequent processing, only quaternion matrices are processed. Quaternion [30] is a mathematical concept that represents super complex numbers. It is composed of the real part plus three virtual parts: $i, j, k$. The following rules are met between the three virtual parts: $i^2 = j^2 = k^2 = ijk = -1$, $ij = -ji = k$, $jk = -kj = I$, $ki = -ik = j$. From here, we can know that each quaternion can be linearly expressed by $1, i, j, k$, then, the general expression of a quaternion is $q = a + b\overleftarrow{i} + c\overleftarrow{j} + d\overleftarrow{k}$. In addition, $a, b, c, d$ is a real number. When $a = 0$, $q = b\overleftarrow{i} + c\overleftarrow{j} + d\overleftarrow{k}$ is called a pure quaternion. In recent years, quaternions have been used widely often in signal processing [31], coordinate solving [32], and image processing [33]. The quaternion represents a color image that avoids the correlation among R, G, and B during the sub-channel processing. The quaternion representation of color images is used generally in the literature [34]. Let the three imaginary components of the quaternion represent the three primary color components of red (R), green (G), and blue (B); additionally, the real part is 0, which can represent each pixel of a color image as a pure quaternion.

$$f(x, y) = f_R(x, y) \cdot i + f_G(x, y) \cdot j + f_B(x, y) \cdot k. \quad (5)$$

Among them, $f_R(x,y)$, $f_G(x,y)$, and $f_B(x,y)$ represent the R, G, and B components of the color image, respectively; and $x, y$ represent the coordinates of the image matrix where the pixel is located. In this way, the color image can be represented by a pure quaternion matrix. Quaternion-based color image processing is used to handle its quaternion matrix. Compared with the traditional sub-channel or the method of transforming to grayscale image and then processing, the quaternion method can better reflect the color image holism, whether for theoretical innovation or practical application, which provides a new direction.

According to the introduction in the previous section, two main factors should be considered when applying the PSO algorithm to image contrast enhancement: (i) Design of a transformation function that will generate a new pixel intensity of the enhanced image from the original image; and (ii) Design of fitness function that checks the quality of the generated

image. These important factors affect the quality of enhanced images and are thoroughly described in the following sections.

**3.2.1 Transformation function.** A contrast function is used to enhance the contrast of the image in the spatial domain. The transform function generates a new intensity for each pixel of the original image and generates an enhanced image. In this paper, a low-illumination image is mapped by a transformation function to obtain a high-contrast image. This article uses the local/ global enhancement transformation proposed in [35] to use local image statistics, such as mean, variance, and global image information. This function is an extended version of the local enhancement function in [36]. Transform is applied to each pixel of the image of size M*N at position $(x, y)$, and the old intensity $f(x, y)$ is mapped to the new intensity value $g(x, y)$. The formula of this transformation function is

$$
\begin{aligned}
g(x, y) &= T(f(x, y)) \\
&= \kappa * (G_m/(\sigma(x, y) + b)) * [f(x, y) - c * m(x, y)] + (m(x, y))^a.
\end{aligned}
\tag{6}
$$

In Eq (6), $\kappa$, $a$, $b$, and $c$ are four parameters, $m(x, y)$ is the local mean of the $(x, y)^{th}$ pixel of the input image over a n×n window. $G_m$ is the global mean, and $\sigma(x, y)$ is the local standard deviation of $(x, y)^{th}$ pixel of the input image over a n×n window, which are defined as follows:

$$
m(x, y) = \frac{1}{n \times n} \sum_{x=0}^{n-1} \sum_{y=0}^{n-1} f(x, y),
\tag{7}
$$

$$
G_m = \frac{1}{M \times N} \sum_{i=0}^{M-1} \sum_{j=0}^{N-1} f(i, j),
\tag{8}
$$

$$
\sigma(i, j) = \sqrt{\frac{1}{n \times n} \sum_{x=0}^{n} \sum_{y=0}^{n} (f(x, y) - m(i, j))^2}.
\tag{9}
$$

Eq (6) has four unknown parameters, i.e., $\kappa$, $a$, $b$, and $c$, and these parameters cause considerable changes to the processed image. It enhances low-contrast images with a local mean enhancement center. The local mean and local standard deviation define the local brightness and contrast of the image, respectively. Parameter *an* introduces smoothness and brightness effects in the image, while $b$ introduces an offset from the standard deviation in the neighborhood. Parameter $c$ controls how much the average value is to be subtracted from the image $f(u, v)$. Finally, parameter $\kappa$ controls the global enhancement of the image. For these four optimization parameters, the parameter search range given by different authors is different, which affects the quality of the enhanced image. In [36], the parameter $b$ is set to zero, which makes the master dependent on the local variance. Therefore, zero local variance causes instability in the conversion. In addition, parameters $a$, $c$, and $\kappa$ are set to 1, which limits the optimal choice of parameters to achieve the best performance. In [22], the parameters are as follows: $a$, $b$, and $c$ can have any true positive non-zero value, and $\kappa$ remains between 0.5 and 1.5. Therefore, the purpose of the optimization algorithm is to determine the values of these parameters on the bases of a given objective function, thereby obtaining the best-enhanced image.

**3.2.2 Fitness function.** As mentioned above, another factor that affects the enhancement of image contrast is the fitness value function. The fitness function is an objective evaluation criterion for automatically measuring the quality of the generated image. Enhancing image quality is the key to determining image quality. Intuitively, compared with the original image, the enhanced image needs other edges, higher edge strength, and higher contrast. For color

images, chroma is a judgment index. On the basis of this fact, a new fitness function is proposed, which contains four performance indicators: (i) the sum of edge strength, (ii) the number of edge pixels, (iii) color entropy, (iv) image contrast, and (v) Chroma. More specifically, given the original image, this method will enhance the image and generate an enhanced image on the basis of the following fitness function. In this paper, the formula is expressed as follows:

$$f(I_e) = \log(\log(E(I_s))) \times ((n\_edgels(I_s))/(M \times N)) \times H(I_e) \times C(I_e) \times I(I_e), \quad (10)$$

Where $I_e$ is the enhanced image of the original image. $I_s$ is an edge image on the produced enhanced image. $E(I_s)$ is the image edge intensity, $H(I_e)$ is the color entropy, $n\_edgels(I_s)$ is the number of image edge pixels, $C(I_e)$ is the image contrast, and $I(I_e)$ is the image brightness.

Image contrast refers to the measurement of different brightness levels between the brightest white and the darkest black in the light and dark areas of an image, that is, the magnitude of the gray contrast of an image. The larger the range of difference, the more vivid and rich colors can be easily displayed. The image contrast calculation formula is

$$C = \sum_{\delta} \delta(i,j)^2 P_\delta(i,j), \quad (11)$$

Where $\delta(i,j) = |i - j|$ is the quaternion difference between adjacent pixels, and $P_\delta(i,j)$ is the pixel distribution probability of quaternion difference between adjacent pixels.

Image entropy is expressed as the average number of bits in the gray level set of the image, and it describes the average amount of information of the image source. The color entropy calculation of the image is to first calculate the image entropy for the three channels and then average the information of the three channels. The image entropy calculation formula is as follows:

$$H = -\sum_{i=0}^{255} p_i \log p_i, \quad (12)$$

where $p_i$ is the probability that a certain gray level appears in the image.

The edge information is obtained by the Sobel [37] detector, and the calculation formula is as follows:

$$
\begin{aligned}
Gx \quad &= (-1) * f(x-1, y-1) + 0 * f(x, y-1) + 1 * f(x+1, y-1) + (-2) * f(x-1, y) + 0 * f(x, y) \\
&+ 2 * f(x+1, y) + (-1) * f(x-1, y+1) + 0 * f(x, y+1) + 1 * f(x+1, y+1) \\
&= [f(x+1, y-1) + 2 * f(x+1, y) + f(x+1, y+1)] - [f(x-1, y-1) + 2 * f(x-1, y) + f(x-1, y+1)]
\end{aligned}
\quad , (13)
$$

$$
\begin{aligned}
Gy \quad &= 1 * f(x-1, y-1) + 2 * f(x, y-1) + 1 * f(x+1, y-1) + 0 * f(x-1, y) + 0 * f(x, y) \\
&+ 0 * f(x+1, y) + (-1) * f(x-1, y+1) + (-2) * f(x, y+1) + (-1) * f(x+1, y+1) \\
&= [f(x-1, y-1) + 2 * f(x, y-1) + f(x+1, y-1)] - [f(x-1, y+1) + 2 * f(x, y+1) + 2 * f(x+1, y+1)]
\end{aligned}
\quad , (14)
$$

where $f(a, b)$ represents the quaternion value of the point $(a, b)$. The horizontal and vertical gray value of each pixel of the image is calculated by the following formula:

$$|G| = |Gx| + |Gy|. \quad (15)$$

Brightness corresponds to imaging brightness and image gray and is the brightness of the color. According to the sensitivity of the human eye to the three primary colors of R, G, and B,

three different coefficients are obtained. The brightness calculation formula is

$$I = 0.299 * R + 0.587 * G + 0.114 * B. \tag{16}$$

**3.2.3 Image contrast enhancement approach.** In this work, the input image is first represented by quaternion, and the image function is used as input. To generate an augmented image from an input image, a parametric transformation function is defined using Eq (6), which combines global and local information of the input image. The transformation function contains four parameters, namely, $\kappa$, a, b, and c. These four parameters have their defined ranges, and different values produce different enhanced images. Our goal is to find the set of values for these four parameters that produce the best results (according to fitness function values) using PSO. Through the steps on image representation, image transformation, and fitness function, this section describes the algorithm steps and advantages for image contrast enhancement on the bases of the improved particle swarm algorithm.

**Initialization**. In this step, using particle swarm-related parameters such as population size, search space range, and velocity, among others, the initial solution is obtained. At the same time, the parameters of $\kappa$, a, b, and c in the transformation function are initialized. First, *P* particles are initialized. The position vector of each particle *X* has four components, namely, k, a, b, and c. Now, using these parameter values, each particle generates an enhanced image using the intensity transformation function defined in Eq (6). The transform function is applied to each pixel in the input image, takes parameter values from each particle, and generates a modified intensity value for that pixel. Thus, each generation produces P enhanced images, and the quality of each enhanced image is measured by the objective function (fitness function) defined in Eq (16).

In the traditional particle swarm algorithm, the fitness values of all enhanced images generated by all particles are calculated. In PSO, the most attractive property is the direction in which PBEST and GBEST are responsible for driving each particle (solution) to the best position. In each iteration, P new positions are generated, and the PBEST and GBEST of each generation are found according to their fitness values. With the help of these two optimal values, the component new velocities of each particle are calculated to obtain the optimal solution used. When the process is complete, the augmented image is created from the GBEST position of the particle as it provides the maximum fitness value.

When using traditional PSO for image enhancement, only two items, namely, PBEST and GBEST, are considered. To further increase the diversity of particles, this paper adds a local optimum for topology calculation to fully utilize the diversity of particles. Therefore, the quaternion representation image is iteratively updated by formula 6 to update the velocity and position of particles. Then, formula 16 calculates the fitness value of the augmented image generated by all particles. In each step of iteration, P new positions will be generated. The individual optimum, global optimum, and local optimum of each generation are found according to the fitness value. When the three are balanced, the obtained optimum value is the global optimum. Fig 2 shows the flowchart of the algorithm.

## 4 Experimental results

This paper verifies the effectiveness of the algorithm via two experiments. First, it simulates and tests the improved particle swarm algorithm, and then, it verifies the convergence performance of the algorithm by standard test functions. Second, for the image enhancement method, the images in the test set are selected and evaluated via subjective vision and objective quality criteria.

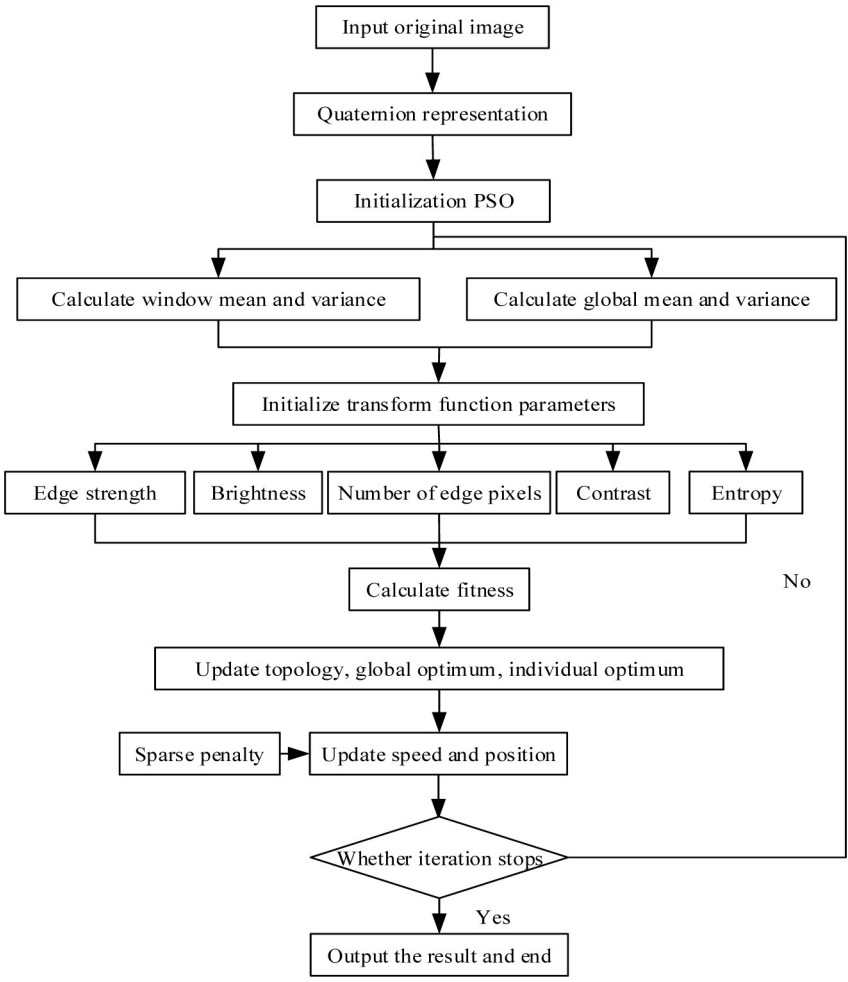

**Fig 2. Flowchart of the algorithm.**

## 4.1 Improved PSO experimental simulation

This paper selects six benchmark test functions (two single-peak functions, two multi-peak functions, and two multi-peak fixed-dimensional test functions) as shown in Table 1 for the algorithm. Verification is performed to determine the minimum value of the function, and the results of benchmark function optimization are compared and analyzed with the basic PSO, EDAPSO [38], and TPSO [39], and the algorithm proposed in this paper. Experimental data are noted to observe the same convergence efficiency and search accuracy of the algorithm under the same test function.

According to the parameters required for the experiment, the parameters are set as follows: the maximum number of iterations is 1000, inertia weight $\omega = 0.8$, contraction factor $\chi = 0.729$, and particle size popsize = 30. The results of the experiment on the improved algorithm are that $c_1$, $c_2$, and $c_3$ represent random numbers that the particle learns from individual optimal, local optimal, and global optimal, respectively. When $c_1 = 1.4$, $c_2 = 1.4$, $c_3 = 1.8$, the effect is better.

To easily check the performance of different algorithms, this research compares the traditional PSO, EDAPSO, TPSO, and the algorithms. Each algorithm is independently run 30 times for the six test functions in Table 1. Table 2 shows the minimum, average, and average

**Table 1. Benchmark functions.**

| Number | Function | Expression | Scope | Optimal |
|---|---|---|---|---|
| F1 | Schwefel problem 1.2 | $$F_1(x) = \sum_{i=1}^{D}\left(\sum_{j=1}^{i} x_j\right)^2$$ | $(-100 \le x_i \le 100)$ | 0 |
| F2 | Schwefel problem 2.2 | $$F_2(x) = \sum_{i=1}^{D}|x_i| + \prod_{i=1}^{D}|x_i|$$ | $(-10 \le x_i \le 10)$ | 0 |
| F3 | Levy | $$F_4(x) = \sin^2(3\pi x_i) + (x_i - 1)^2\left[1 + \sin^2(3\pi x_{i+1})\right] + (x_{i+1} - 1)^2\left[1 + \sin^2(2\pi x_{i+1})\right]$$ | $(-10 \le x_i \le 10)$ | 0 |
| F4 | Ackley | $$F_3(x) = 20 + \exp - 20\exp\left(-0.2\sqrt{\frac{1}{n}\sum_{i=1}^{n}x_i^{2}}\right) - \exp\left(\frac{1}{n}\sum_{i=1}^{n}\cos(2\pi x_i)\right)$$ | $(-30 \le x_i \le 30)$ | 0 |
| F5 | Three-hump Camel | $$F_5(x) = 4x_1^2 - 2.1x_1^2 + \tfrac{1}{3}x_1^6 + x_1 x_2 - 4x_2^2 + 4x_2^4$$ | $(-5 \le x_i \le 5)$ | -1.0316 |
| F6 | Goldstein price | $$F_6(x) = \left[1 + (x_1 + x_2 + 1)^2(19 - 14x_1 + 3x_1^2 - 14x_2 + 6x_1 x_2 + 3x_2^2)\right] \times \left[30 + (2x_1 - 3x_2)^2(18 - 32x_1 + 12x_1^2 + 48x_2 - 36x_1 x_2 + 27x_2^2)\right]$$ | $(-2 \le x_i \le 2)$ | 3 |

times. By comparing the data in Table 2, the improved PSO algorithm has obvious advantages over the other three PSO, EDAPSO, and TPSO. Considering that the local optimal value is added to the algorithm in this paper and comparing the data in the table, the algorithm in this paper can converge to the global optimal value, especially in F3, F5, and F6, after the

**Table 2. Function test results.**

| Function | Algorithm | Max | Min | Mean | Average time (s) |
|---|---|---|---|---|---|
| F1 | PSO | 4.71E+06 | 7.25E+03 | 4.42E+04 | 2.8713 |
|  | EDAPSO | 1.09E+06 | 1.40E+03 | 7.94E+03 | 7.3747 |
|  | TPSO | 8.01E+05 | 2.2E-177 | 4.85E+03 | 2.5738 |
|  | proposed | 2.68E+05 | **3E-183** | **1.24E+03** | 2.5733 |
| F2 | PSO | 2.28E+55 | 1.46E+02 | 2.32E+52 | 2.4711 |
|  | EDAPSO | 7.48E+37 | 53.8 | 1.01E+35 | 6.5833 |
|  | TPSO | 1.65E+54 | 3.67E-88 | 1.65E+51 | 2.1842 |
|  | proposed | 3.87E+24 | **9.53E-94** | **3.89E+21** | 2.2709 |
| F3 | PSO | 4.02E+03 | 60.36 | 84.49 | 1.5473 |
|  | EDAPSO | 1.45E+03 | 11.38 | 18.53 | 6.2869 |
|  | TPSO | 6.20E+03 | 1.63 | 19.09 | 1.9477 |
|  | proposed | 2.02E+03 | **0** | **0** | 1.8295 |
| F4 | PSO | 20.48 | 7.74 | 8.07 | 1.8511 |
|  | EDAPSO | 19.35 | 2.74 | 3.03 | 4.8946 |
|  | TPSO | 20.73 | **8.88E-16** | 0.27 | 1.5838 |
|  | proposed | 20.38 | **8.88E-16** | **0.17** | 1.3921 |
| F5 | PSO | 7.71E+02 | 1.1E-188 | 1.16 | 1.7388 |
|  | EDAPSO | 7.74E+02 | **0** | 0.86 | 4.8756 |
|  | TPSO | 1.57E+02 | 1.86E-67 | 0.20 | 1.5622 |
|  | proposed | 21.46 | **0** | **0.04** | 1.5728 |
| F6 | PSO | 6.02E+08 | **3** | 6.64E+05 | 1.3622 |
|  | EDAPSO | 6.05E+07 | **3** | 1.82E+05 | 4.6748 |
|  | TPSO | 1.97E+08 | **3** | 2.14E+05 | 1.4667 |
|  | proposed | 1.57E+04 | **3** | **3.61E+01** | 1.3553 |

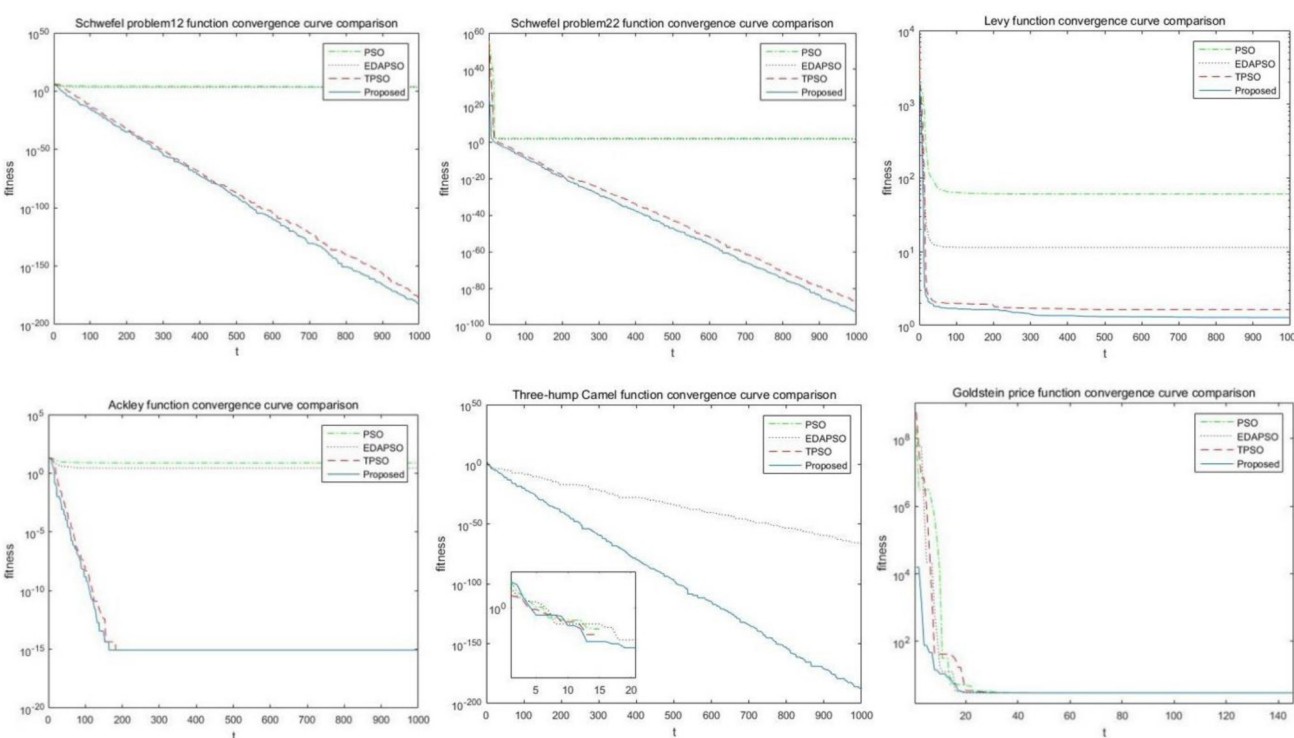

**Fig 3. Comparison of convergence curves of six different test functions.**

comparison of the minimum value and the optimal value of the function. For the single-peak function, comparing the average values of several algorithms, the average value can be concluded to have increased by 4–10 times. For the Schwefel problem 2.2 functions, the average value of convergence has been improved qualitatively, which means the improved algorithm in this paper improved the search accuracy higher than the other three algorithms, and the diversity of particles is fully utilized. For multi-peak functions, a large number of local extremums are included in the entire function search space. The PSO algorithm can easily cause it to fall into local extremums. However, the improved PSO uses the comparison of topological structures to obtain the diversity of particles. Full use makes the algorithm reach a balance in the learning process and enables it to stand out from the local optimal value, allowing it to continue searching for the global optimal value. For the multimodal fixed-dimensional test function, the solutions obtained by different algorithms have little difference due to its low solution space dimension. Although no true value was found, the improved algorithm in this paper generates better search results.

Fig 3 shows the convergence curves of several algorithms applied to the same constraint function. Evidently, the improved PSO algorithm has better convergence than the other three algorithms.

Given that this paper uses a deep topology, it has a certain impact on the time complexity of the entire algorithm. Owing to the addition of a sparse penalty term, the final time complexity presents a middle level. It carries out theoretical analysis and comparison of the average computational time complexity of the proposed algorithm and the traditional PSO algorithm. According to the description of the two algorithms, for the traditional PSO algorithm, the number of particles in each iteration is unchanged. Suppose the number of particles in the $i$-th

iteration is $N_i$, where $i = 1, 2,\ldots,m$, where $m$ represents the maximum number of iterations; thus, $N_1 = N_2 = \ldots = N_m = N$. Assuming that the computing time required for each iteration of each particle is $T_T$, the total running time required by the traditional PSO algorithm for optimization can be concluded to be $N \times m \times T_T$. For the algorithm in this paper, the number of particles slowly decreases with the iteration, that is, $N_1 \geq N_2 \ldots \geq N_m$. Assuming that the computing time required for each iteration of each particle is $T_D$, the total running time required

by the algorithm in this paper after optimization is $\sum_{i}^{m} N_i \times T_D$. From the above analysis, the

difference in the complexity of the two algorithms is reflected in the number of particles in each iteration and the running time required for each iteration of each particle.

For the traditional PSO algorithm, the operations required for a particle to update are as follows: 5 multiplications and 5 additions. The times required for multiplication and addition operations are assumed to be $T_m$ and $T_a$. In the experiment, the number of particles in the traditional PSO algorithm is set to 50, and the number of iterations is $m$. Therefore, the average time required for the traditional PSO algorithm to complete the optimization is $N \times m \times T_T = m \times 50 \times (5T_m + 5T_a)$. For the algorithm in this paper, the operations required for a particle to compare and update are as follows: $\frac{5}{4^n} T_m (n = 1, 2, \ldots, m)$ multiplications and $\frac{5}{4^n} T_a (n = 1, 2, \ldots, \text{m})$ additions. In the experiment, the maximum number of iterations is set to 1,000. Therefore, the average time required for the algorithm in this paper to complete the

optimization is $\sum_{i=1}^{m} N_i \times T_D = \sum_{i=1}^{m} N_i \circ \left( \frac{5}{4^n} T_m + \frac{5}{4^n} T_a \right) n = 1, 2, \ldots, m. \ i = 1, 2, \ldots, 1000,$

where $N_i$ represents the number of particles in the $i$-th iteration. When $n$ progressively enlarges, $\frac{5}{4^n}$ continuously diminishes. After adding the sparse penalty coefficient, the sub-optimal solution is penalized, and the average time of the algorithm in this paper tends to $N \times m \times T_T = m \times 50 \times (T_m + T_a)$. Given that the number of particles in each iteration of the algorithm in this paper is different, conducting multiple experiments, counting the number of particles in the corresponding iterations, and calculating the average number of particles in the corresponding iterations are necessary.

## 4.2 Algorithm performance evaluation

**4.2.1 Subjective evaluation of image quality.** According to the description in [40], the evaluation methods of image quality can be divided into two categories: subjective and objective. The only subjective evaluation is the Mean Opinion Score (MOS), which means that many people judge the quality of an image, and the average of their scores is used as the evaluation result. We use the Mean Opinion Score (MOS) to reinforce the assumption that our proposed contrast enhancement algorithm can better match the human visual system's direct observations of the scene. Subjective experiments were conducted by 30 non-professional audience members who analyzed the color, contrast, sharpness, and texture of the fused images and were usually divided into two categories: absolute evaluation and relative evaluation. Absolute evaluation means that the scorer observes the image and provides the score according to the direct perception of the image. Table 3 lists the 5-level absolute scales listed in the relevant international documents, which are divided into obstructive and quality scales. Obstruction scales are mostly used by professional graders, while quality scales are more widely used. The relative evaluation is to let the raters observe a batch of images, classify them from good to bad, and provide the corresponding scores.

Table 4 is a comparison table of the absolute and relative evaluation scales. Observers rated their quality from 5 (excellent) to 1 (very poor).

**Table 3. Absolute measure.**

| Grade | hinder scale evaluation | Quality scale evaluation |
|---|---|---|
| 5 | No visible deterioration in image quality | Excellent |
| 4 | Changes in image quality can be seen without hindering viewing | Good |
| 3 | It is clearly seen that the image quality has deteriorated, which is slightly obstructive to viewing | Fair |
| 2 | obstructing viewing | Poor |
| 1 | very serious hindrance to viewing | Very poor |

**4.2.2 Objective evaluation of image quality.** In objective image quality evaluation, the main evaluation indicators include peak signal-to-noise ratio (PSNR), structural similarity index measure (SSIM) [41], information fidelity criterion (IFC) [42], visual information fidelity (VIF) [43], feature similarity (FSIM) [44], Information Entropy (Entropy), and Average Gradient (AG). Several evaluation indicators are thoroughly described below:

(1) PSNR is generally used for an engineering project between maximum signal and background noise. To measure the image quality after processing, the PSNR value is generally used to measure whether a processing program is satisfactory. It is the logarithm of the mean square error between the original image and the processed image with respect to $(2^n - 1)^2$ (the square of the maximum value of the signal, n is the number of bits per sample), and its unit is dB. PSNR is defined by the mean square error (MSE), two images $I$ and $K$, sizes are M × N, if one is the noise approximation of the other; the mean square error is defined as follows:

$$MSE = \frac{1}{M \times N} \sum_{i=0}^{M-1} \sum_{j=0}^{N-1} ||I(i,j) - K(i,j)||^2 \tag{17}$$

Therefore, the PSNR is defined as

$$PSNR = 10 \cdot \log_{10}\left(\frac{(2^n - 1)^2}{MSE}\right) = 10 \cdot \log_{10}\left(\frac{MAX_I^2}{MSE}\right) = 20 \cdot \log_{10}\left(\frac{MAX_I}{\sqrt{MSE}}\right) \tag{18}$$

where $MAX_I$ represents the maximum value of the image point color.

(2) SSIM. The human visual system can highly adaptively extract the structural information in the scene. Considering the distortion of the image by comparing the changes in the image structure information, the objective quality evaluation SSIM is obtained. The evaluation model of SSIM is as follows:

$$SSIM(x,y) = \frac{\left(2u_x u_y + c_1\right)\left(2\sigma_{xy} + c_2\right)}{\left(u_x^2 + u_y^2 + c_1\right)\left(\sigma_x^2 + \sigma_y^2 + c_2\right)} \tag{19}$$

**Table 4. Subjective method for image quality assessment.**

| Grade | Absolute measure | Relative measure |
|---|---|---|
| 5 | Excellent | The best in the group |
| 4 | Good | Better than the average level in the group |
| 3 | Fair | Average level in the group |
| 2 | Poor | Lower than the average level |
| 1 | Very poor | Lowest in the group |

where $x$ and $y$ are the reference and images to be tested, and $\mu_x, \mu_y, \sigma_x^2, \sigma_y^2, \sigma_{xy}$ is the mean, variance, and covariance of the images $x$ and $y$. $C_1, C_2$ is a small positive number to avoid instability due to zero denominator in the above formula. The closer $x$ is to $y$, the closer the value of $SSIM(x, y)$ is to 1. The closer the $SSIM$ is to 1, the better the quality of the image.

(3) IFC. According to the IFC standard, the mutual information between the content observed by the human eye and the content of the image itself is assumed to be positively correlated, that is, the mutual information between the image information observed by the human eye and the content of the image itself characterizes the content observed by the human eye. When the image is distorted, the amount of mutual information between the observed and original images will decrease. If we can calculate the mutual information of the original reference image and the distorted image observed by the human eye and then calculate the mutual information of the original reference image and the reference image observed by the human eye, we can calculate the proportion of the information of the distorted image. The ratio of the amount of reference image information that the human eye can accept and this ratio can also characterize the image quality of the distorted image, which is the index we ultimately require.

(4) VIF. Starting from the knowledge of information theory, the distorted image is regarded as a decrease in information fidelity. The common information of the reference image and the test image is correlated, that is, the mutual information of the two is correlated. Mutual information is a statistical measure of information fidelity. Although it is not strongly correlated with image quality, it can limit the extraction of cognitive information from images. Information fidelity explores the relationship between visual quality and image information. Association. In the VIF evaluation model, the input image, the image distortion channel, and the distorted image model are all assumed to be accurate. The VIF indicator can be expressed as

$$VIF = \frac{\sum_{k=1}^{K} \left[ I\left( C_r^k; F^k | z_r^k \right) \right]}{\sum_{k=1}^{K} \left[ I\left( C_r^k; E^k | z_r^k \right) \right]} \tag{20}$$

$I\left( C_r^k; E^k | z_r^k \right)$ and $I\left( C_r^k; F^k | z_r^k \right)$ are the corresponding mutual information measurements of the $k$-th sub-band, respectively, where $k$ is the number of sub-bands.

(5) FSIM. FSIM is an image evaluation index based on the underlying features. It uses phase consistency to extract the underlying features of the image, which is closer to the human visual system. The main calculation methods are

$$FSIM = \frac{S_{PC}(x, y) \cdot S_G(x, y) \cdot PC_m(x, y)}{\sum_{x,y \in \Omega} PC_m(x, y)} \tag{21}$$

$$S_{PC}(x, y) = \frac{2PC(x) \cdot PC(y) + T_1}{PC^2(x) + PC^2(y) + T_1} \tag{22}$$

$$S_G(x, y) = \frac{2G(x) \cdot G(y) + T_2}{G^2(x) + G^2(y) + T_2} \tag{23}$$

where $PC_m(x, y) = \max(PC(x), PC(y))$ is used to weight the overall similarity of the image $x$ and $y$. $S_{PC}(x, y)$ represents the feature similarity of images $x$ and $y$, $S_G(x, y)$ represents the gradient similarity, $PC$ represents the phase consistency information, and $G$ represents the gradient

amplitude of the image. The constant is introduced to avoid the situation where the denominator is zero in the above format. The larger the value of FSIM, the more similar the reference image and the image to be tested and the higher the quality of the image to be tested, and vice versa, and the lower the quality of the image to be tested.

(6) AG. The average gradient can reflect the detailed contrast and texture transformation in the image. It somewhat reflects the clarity of the image. The calculation formula is

$$G = \frac{1}{M \times N} \sum_{i=1}^{M} \sum_{j=1}^{N} \sqrt{\frac{\left(\frac{\partial f}{\partial x}\right)^2 + \left(\frac{\partial f}{\partial y}\right)^2}{2}} \tag{24}$$

M×N represents the size of the image, $\frac{\partial f}{\partial x}$ represents the gradient in the horizontal direction, and $\frac{\partial f}{\partial y}$ represents the gradient in the vertical direction.

**4.2.3 Results and analysis.** This study selected six images from two test image sets, and it performs contrast enhancement on different algorithms. In order to reflect the effectiveness of the algorithm in this paper, the images used in this paper are images from other datasets and network images as a reference, and we collect multiple low-contrast images in different scenes. The figure shows the effect of the enhancement. This paper selects six images from two data sets as reference images and compares them with other seven types on the bases of evolution. Contrast image enhancement methods were compared, including the following: based on artificial bee colony algorithm, cuckoo search algorithm, scoring algorithm, butterfly algorithm, genetic algorithm, and basic particle swarm algorithm; and performance evaluation using 7-image quality criteria. All these criteria are widely used in the field of image contrast enhancement and evaluate the performance of various image contrast enhancement methods. The parameters of all methods set the maximum number of iterations at 50, and the swarm of population size is set to be 50. All test images use the same parameter settings because the proposed method is quite robust to the choice of parameters used.

Fig 4 shows a comparison of the effects of the input images and the enhanced images. Tables 5–8 contain the objective index data of the enhanced images obtained by the six test images according to different algorithms. Each table corresponds to the comparative data of different performance indexes of an image under the seven algorithms. The values in entropy and AG brackets are for the original image. The diagrams and tables show that this method is superior to other optimization algorithms for image enhancement methods and achieves the best image visual quality and the best target performance.

Table 9 shows the calculation time of the seven algorithms for different images. The table clearly shows that the running time of the proposed algorithm is improved qualitatively compared with other algorithms. The optimal value is shown in bold.

Table 10 shows the MOS scores of Figs 1–4 under several image enhancement methods. By comparing the MOS of several algorithms, the results show that our method achieves the highest MOS value on all enhanced images, which indicates that the method is visually superior to other state-of-the-art algorithms.

Another experiment was conducted to compare the proposed approach with a few recent image contrast enhancement approaches using selected images. Fig 5 presents the input images and enhancement images. Tables 11–14 are the objective index data of the enhanced images obtained by the four test images according to different algorithms.

Tables 11–14 correspond to the comparative data of different performance indexes of an image under the four algorithms, where one can see that the proposed approach can enhance the image contrast. Table 15 shows the calculation time of the four algorithms for different

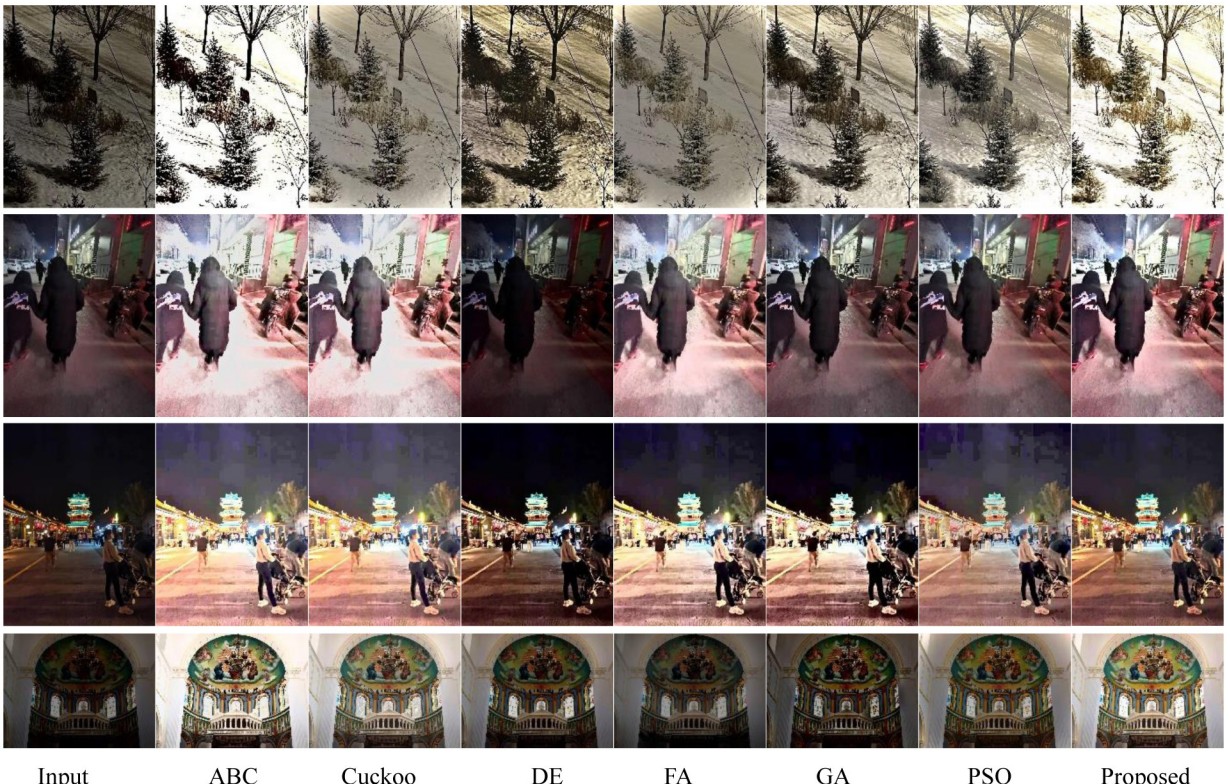

|  | Input | ABC | Cuckoo | DE | FA | GA | PSO | Proposed |

**Fig 4. The visual quality performance comparison of the enhanced image obtained by various image enhancement approaches.** The first column is the input images, the second to the eighth column are results of ABC, Cuckoo, DE, FA, GA, PSO, respectively. The last column is the results of the proposed approach. From top to bottom, the test images are images 1–4, respectively.

**Table 5. The performance evaluation of various image contrast enhancement approaches using test images 1.**

| Image1 | method | PSNR | SSIM | IFC | VIF | FSIM | Entropy (6.3489) | AG(0.0482) |
|---|---|---|---|---|---|---|---|---|
|  | ABC | 39.9824 | 0.5827 | 1.0283 | 0.3018 | 0.5766 | 5.8493 | 0.0885 |
|  | Cuckoo | 38.5637 | 0.5092 | 0.0801 | 0.2603 | 0.5239 | 5.4863 | 0.0772 |
|  | DE | 35.4092 | 0.4222 | 0.6748 | 0.1987 | 0.4783 | 5.1192 | 0.0575 |
|  | FA | 38.8869 | 0.5181 | 0.0826 | 0.2756 | 0.5187 | 5.4958 | 0.0767 |
|  | GA | 39.4758 | 0.5746 | 0.0938 | 0.2981 | 0.5837 | 5.6475 | 0.0866 |
|  | PSO | 40.8482 | 0.6326 | 1.7463 | 0.2983 | 0.6018 | 6.2959 | 0.1022 |
|  | Proposed | **48.0932** | **0.7827** | **2.1182** | **0.3749** | **0.7633** | **7.2739** | **0.1673** |

**Table 6. The performance evaluation of various image contrast enhancement approaches using test images 2.**

| Image2 | method | PSNR | SSIM | IFC | VIF | FSIM | Entropy(3.4982) | AG(0.0139) |
|---|---|---|---|---|---|---|---|---|
|  | ABC | 26.4348 | 0.5374 | 0.8498 | 0.4096 | 0.5557 | 4.6891 | 0.0278 |
|  | Cuckoo | 26.4994 | 0.5371 | 0.8526 | 0.4721 | 0.5623 | 4.9819 | 0.0309 |
|  | DE | 23.7586 | 0.3219 | 0.6875 | 0.4759 | 0.4196 | 3.5781 | 0.0176 |
|  | FA | 27.4373 | 0.5777 | 0.8421 | 0.4127 | 0.5684 | 5.0215 | 0.0363 |
|  | GA | 28.3584 | 0.6365 | 1.2382 | 0.4365 | 0.6232 | 5.2819 | 0.0428 |
|  | PSO | 29.8473 | 0.6565 | 1.2788 | 0.4881 | 0.6758 | 5.5611 | 0.0683 |
|  | Proposed | **37.6251** | **0.8175** | **2.7586** | **0.8571** | **0.8039** | **6.2967** | **0.0849** |

**Table 7. The performance evaluation of various image contrast enhancement approaches using test images 3.**

| Image3 | method | PSNR | SSIM | IFC | VIF | FSIM | Entropy(4.9283) | AG(0.0175) |
|--------|--------|------|------|-----|-----|------|-----------------|------------|
| | ABC | 26.9891 | 0.4375 | 0.7463 | 0.1278 | 0.3718 | 3.8758 | 0.0184 |
| | Cuckoo | 28.8284 | 0.5123 | 1.0632 | 0.2920 | 0.4921 | 4.6274 | 0.0283 |
| | DE | 29.3456 | 0.5351 | 1.3372 | 0.3162 | 0.5291 | 4.9034 | 0.0337 |
| | FA | 28.9483 | 0.5134 | 0.9744 | 0.3093 | 0.5018 | 4.7683 | 0.0237 |
| | GA | 30.6166 | 0.5788 | 1.5937 | 0.3592 | 0.5621 | 5.1627 | 0.0472 |
| | PSO | 31.8593 | 0.6123 | 1.6281 | 0.3626 | 0.5901 | 5.0192 | 0.0589 |
| | Proposed | **35.9381** | **0.6776** | **2.6887** | **0.6471** | **0.6827** | **5.6572** | **0.0773** |

**Table 8. The performance evaluation of various image contrast enhancement approaches using test images 4.**

| Image4 | method | PSNR | SSIM | IFC | VIF | FSIM | Entropy(4.826) | AG(0.0313) |
|--------|--------|------|------|-----|-----|------|----------------|------------|
| | ABC | 46.4861 | 0.6263 | 2.1232 | 0.2029 | 0.8174 | 5.4322 | 0.0559 |
| | Cuckoo | 47.1777 | 0.6182 | 2.3693 | 0.1399 | 0.7802 | 5.6528 | 0.0566 |
| | DE | 46.2921 | 0.6233 | 1.9447 | 0.3192 | 0.8600 | 5.4792 | 0.0512 |
| | FA | 45.9491 | 0.5588 | 0.8372 | 0.1017 | 0.7954 | 5.6351 | 0.0571 |
| | GA | 40.5044 | 0.4435 | 0.7879 | 0.0970 | 0.7558 | 5.3736 | 0.0387 |
| | PSO | 46.6007 | 0.5505 | 1.3989 | 0.1549 | 0.8220 | 5.3837 | 0.0598 |
| | Proposed | **49.6988** | **0.6907** | **2.9784** | **0.3922** | **0.8633** | **7.5021** | **0.0858** |

**Table 9. The execution time performance (in s) evaluation of various image contrast enhancement approaches.**

| Image | ABC | Cuckoo | DE | FA | GA | PSO | Proposed |
|-------|-----|--------|-----|-----|-----|-----|----------|
| 1 | 413.4708 | 152.4987 | 56.3841 | 87.1932 | 446.3842 | 109.0803 | 45.8375 |
| 2 | 420.3819 | 165.2837 | 60.3716 | 97.3724 | 461.8472 | 116.7566 | 48.7282 |
| 3 | 427.6582 | 163.0495 | 68.0911 | 96.9382 | 472.4098 | 120.8473 | 50.7364 |
| 4 | 409.4721 | 150.6126 | 49.3273 | 85.0674 | 434.8549 | 107.6572 | 41.0843 |

**Table 10. Comparison of various contrast enhancement techniques using MOS.**

| Image | ABC | Cuckoo | DE | FA | GA | PSO | Proposed |
|-------|-----|--------|-----|-----|-----|-----|----------|
| 1 | 56 | 45 | 40 | 48 | 52 | 65 | **73** |
| 2 | 41 | 46 | 37 | 50 | 54 | 55 | **61** |
| 3 | 36 | 42 | 48 | 45 | 51 | 53 | **60** |
| 4 | 62 | 65 | 52 | 51 | 50 | 56 | **76** |

images. The table shows that the running time of the proposed algorithm is improved qualitatively compared with other algorithms.

Table 16 shows the MOS scores of images 5–8 under several image enhancement methods. By comparing the MOS of several algorithms, the results show that our method achieves the highest MOS value on all enhanced images, which indicates that the method is visually superior to other state-of-the-art algorithms.

## 5 Conclusions

With the aim to address the issue of long processing time and poor quality of low contrast enhancement by evolutionary algorithm optimization, this paper proposes an improved

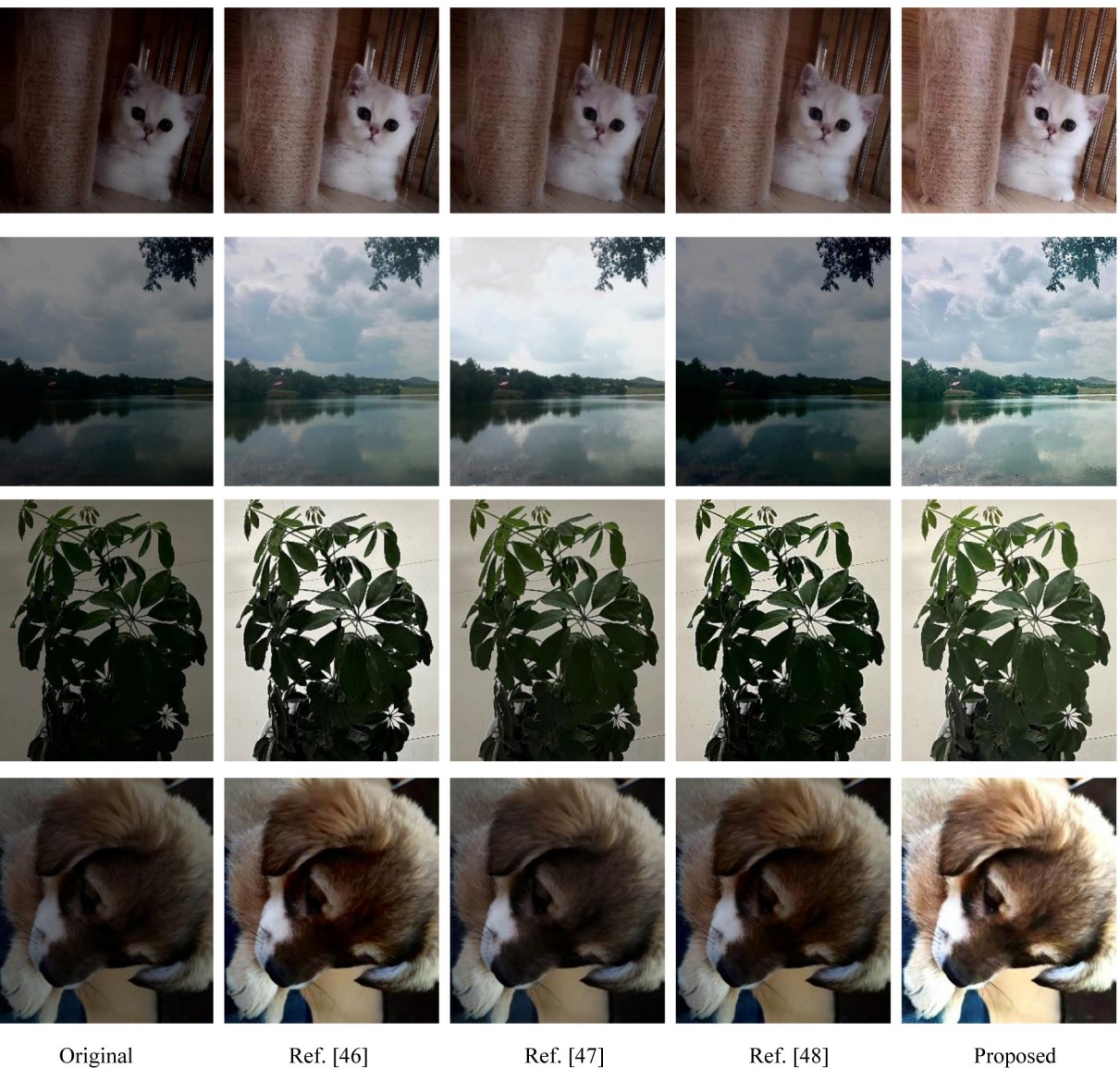

| Original | Ref. [46] | Ref. [47] | Ref. [48] | Proposed |

**Fig 5. Visual quality performance comparison of the enhanced image obtained by various image enhancement approaches.** The first column is the input images, the second to the fourth column2 are results of Ref. [45–47], respectively. The last column is the results of the proposed approach. From top to bottom, the test images are image 5–8.

**Table 11. The performance evaluation of various image contrast enhancement approaches using test images 5.**

| Image5 | method | PSNR | SSIM | IFC | VIF | FSIM | Entropy(5.3818) | AG(0.0349) |
|---|---|---|---|---|---|---|---|---|
| | Ref. [45] | 32.0512 | 0.7785 | 2.1843 | 0.3237 | 0.8689 | 5.8721 | 0.0381 |
| | Ref. [46] | 33.8049 | 0.8306 | 3.1103 | 0.4003 | 0.8853 | 5.7192 | 0.0424 |
| | Ref. [47] | 31.5446 | 0.9136 | 4.0441 | 0.5123 | 0.9347 | 5.1778 | 0.0368 |
| | Proposed | **36.2669** | **0.9470** | **6.1938** | **1.0424** | **0.9572** | **6.1659** | **0.0771** |

**Table 12. The performance evaluation of various image contrast enhancement approaches using test images 6.**

| Image6 | method | PSNR | SSIM | IFC | VIF | FSIM | Entropy(5.2933) | AG(0.0165) |
|---|---|---|---|---|---|---|---|---|
| | Ref. [45] | 45.8738 | 0.4284 | 0.6855 | 0.3647 | 0.6744 | 6.3175 | 0.0196 |
| | Ref. [46] | 46.6472 | 0.4381 | 0.7101 | 0.3811 | 0.6829 | 6.3386 | 0.0187 |
| | Ref. [47] | 47.1637 | 0.4309 | 0.6983 | 0.3796 | 0.6799 | 6.8191 | 0.0206 |
| | Proposed | **56.9928** | **0.5047** | **1.5363** | **0.5029** | **0.7811** | **7.7366** | **0.0386** |

**Table 13. The performance evaluation of various image contrast enhancement approaches using test images 7.**

| Image7 | method | PSNR | SSIM | IFC | VIF | FSIM | Entropy(4.0192) | AG(0.0167) |
|---|---|---|---|---|---|---|---|---|
| | Ref. [45] | 38.3788 | 0.3562 | 1.5919 | 0.5723 | 0.6646 | 6.2379 | 0.0326 |
| | Ref. [46] | 36.3849 | 0.3274 | 0.6728 | 0.4894 | 0.6372 | 5.9283 | 0.0247 |
| | Ref. [47] | 39.5822 | 0.3589 | 1.8473 | 0.6029 | 0.6742 | 6.5674 | 0.0384 |
| | Proposed | **46.9409** | **0.5301** | **3.0571** | **1.1733** | **0.8918** | **7.4985** | **0.0476** |

**Table 14. The performance evaluation of various image contrast enhancement approaches using test images 8.**

| Image8 | method | PSNR | SSIM | IFC | VIF | FSIM | Entropy(6.4021) | AG(0.0913) |
|---|---|---|---|---|---|---|---|---|
| | Ref. [45] | 38.5748 | 0.3562 | 1.4753 | 0.1428 | 0.7636 | 6.8503 | 0.1452 |
| | Ref. [46] | 36.1924 | 0.3109 | 0.9856 | 0.1059 | 0.7142 | 6.4211 | 0.1111 |
| | Ref. [47] | 37.1627 | 0.3322 | 1.1866 | 0.1321 | 0.7376 | 6.4327 | 0.1287 |
| | Proposed | **44.4738** | **0.6274** | **2.5831** | **0.3728** | **0.9103** | **7.4895** | **0.1771** |

**Table 15. The execution time performance (in s) evaluation of various image contrast enhancement approaches.**

| Image | Ref. [45] | Ref. [46] | Ref. [47] | Proposed |
|---|---|---|---|---|
| 1 | 758.29 | 753.25 | 695.33 | 451.06 |
| 2 | 732.65 | 740.93 | 684.32 | 427.57 |
| 3 | 746.01 | 748.72 | 687.29 | 439.64 |
| 4 | 767.39 | 773.48 | 703.66 | 463.21 |

**Table 16. Comparison of various contrast enhancement techniques using MOS.**

| Image | Ref. [45] | Ref. [46] | Ref. [47] | Proposed |
|---|---|---|---|---|
| 5 | 60 | 55 | 57 | **78** |
| 6 | 58 | 55 | 52 | **81** |
| 7 | 62 | 56 | 63 | **73** |
| 8 | 65 | 53 | 60 | **82** |

particle swarm algorithm to optimize the low contrast color image enhancement algorithm. This algorithm used the quaternion matrix to represent the three channels of RGB images. The matrix can perform function transformation and parameter optimization. In the parameter optimization stage, an improved particle swarm algorithm is used. In the improved algorithm, the individual optimization of the particles, local optimization, and global optimization are used to adjust the particle's flight direction. Local optimization uses a topological structure,

and the particles communicate with each other. The local optimal solution is obtained, which not only increases the diversity of particles but also prevents subsequent particles from stopping iteration because they are trapped in local optimization. At the same time, a sparse penalty term is added to the speed update formula. The function of this term is to eliminate the least solution and reduce the iteration time. In the contrast enhancement process, the transformation function and the fitness value function are used to change the image quality. The algorithm in this paper adds contrast elements and brightness elements to the fitness function and uses the fitness function in guiding the particle swarm algorithm to optimize the four parameters in the transformation function. This article compares the proposed algorithm with other evolutionary algorithms to optimize the contrast enhancement. The images in two different data sets are selected for subjective and objective evaluation. The result implies the superiority of the effect in both subjective-qualitative and objective quantitative aspects.

## Supporting information

**S1 Table. Relevant data underlying the findings described in Table 2.**
(XLSX)

**S2 Table. Relevant data underlying the findings described in Tables 10 and 16.**
(XLSX)

**S3 Table. Relevant data underlying the findings described in Tables 5–9 and 11–15.**
(XLSX)

**S1 Fig. Relevant data underlying the findings described in Figs 4, 5.**
(RAR)

## Author Contributions

**Data curation:** Yongfeng Ren.

**Formal analysis:** Guoyong Zhen.

**Investigation:** Yanhu Shan.

**Methodology:** Chengqun Chu.

**Writing – original draft:** Xiaowen Zhang.

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
