## [Decision Letter · Decision Letter 0]

7 Jan 2022

PONE-D-21-36731A color image contrast enhancement method based on improved PSOPLOS ONE

Dear Dr. Ren,

Thank you for submitting your manuscript to PLOS ONE. After careful consideration, we feel that it has merit but does not fully meet PLOS ONE’s publication criteria as it currently stands. Therefore, we invite you to submit a revised version of the manuscript that addresses the points raised during the review process.

ACADEMIC EDITOR:The article require some major corrections in the methodology and experiments.

We look forward to receiving your revised manuscript.

Kind regards,

Diego Oliva

Academic Editor

PLOS ONE

Journal Requirements:

● A clean copy of the edited manuscript (uploaded as the new *manuscript* file).

3. Please clarify in the Methods section how the data were obtained, in enough detail for another researcher to reproduce the analysis, by providing the origin of the images used for testing.

[This work was supported by the National Nature Science Foundation of China (6177012376).]

 [NO]

6. Thank you for stating the following in your Competing Interests section:  

[NO authors have competing interests]. 

7. PLOS requires an ORCID iD for the corresponding author in Editorial Manager on papers submitted after December 6th, 2016. Please ensure that you have an ORCID iD and that it is validated in Editorial Manager. To do this, go to ‘Update my Information’ (in the upper left-hand corner of the main menu), and click on the Fetch/Validate link next to the ORCID field. This will take you to the ORCID site and allow you to create a new iD or authenticate a pre-existing iD in Editorial Manager. Please see the following video for instructions on linking an ORCID iD to your Editorial Manager account: https://www.youtube.com/watch?v=_xcclfuvtxQ.

8. We note that Figures 4 and 5 in your submission contain copyrighted images. All PLOS content is published under the Creative Commons Attribution License (CC BY 4.0), which means that the manuscript, images, and Supporting Information files will be freely available online, and any third party is permitted to access, download, copy, distribute, and use these materials in any way, even commercially, with proper attribution. For more information, see our copyright guidelines: http://journals.plos.org/plosone/s/licenses-and-copyright.

a) You may seek permission from the original copyright holder of Figures 4 and 5 to publish the content specifically under the CC BY 4.0 license. 

Reviewers' comments:

Reviewer's Responses to Questions

**Comments to the Author**

1. Is the manuscript technically sound, and do the data support the conclusions?

Reviewer #1: Yes

Reviewer #2: Partly

2. Has the statistical analysis been performed appropriately and rigorously? 

Reviewer #1: No

Reviewer #2: No

3. Have the authors made all data underlying the findings in their manuscript fully available?

Reviewer #1: Yes

Reviewer #2: Yes

4. Is the manuscript presented in an intelligible fashion and written in standard English?

Reviewer #1: Yes

Reviewer #2: Yes

5. Review Comments to the Author

Reviewer #1: Comments:

1. The authors need to include the quality metrics in abstract to give a clear picture about quality of the proposed work

2. What is the novelty of the proposed PSO compared with existing PSO

3. Need to separate introduction part and related work (literature survey part)

4. The author is expected to include a separate section for to discuss the metrics

PSNR SSIM IFC VIF FSIM

5. Measure the efficiency of the work using qualitative metrics such as Mean Opinion Score

for reference kindly refer the following papers

1. Swarm intelligent based contrast enhancement algorithm with improved visual perception for color images

2. Particle swarm optimisation aided weighted averaging fusion strategy for CT and MRI medical images

3. An optimal weighted averaging fusion strategy for remotely sensed images

4. Optimal fusion aided face recognition from visible and thermal face images

Reviewer #2: The manuscript proposes a color image contrast enhancement method based on improved PSO.

1. The language and presentation need to proof-read and refined.

2. There is no novelty in the transformation function, fitness function etc. It is the same as in Ref [1] The improved PSO algorithm is made use to optimize the selection of parameters k,a,b,c. Authors did not provide any clear mathematical, analytical or theoretical proof to substantiate how the improved version could perform much better as compared with conventional PSO.

3. Comparisons with recent methods are missing. Include comparisons with few state-of-the art non-metaheuristic algorithms as well.

4. Details about the dataset used is missing.

5. Since a deep layered structure was used in the improved variant of PSO proposed, quantify the computational complexity of the entire algorithm

6. The deep layered structure of proposed algorithm is unclear. How are the parameters included in it tuned and selected? Are they image dependent?

7. Since metaheuristic algorithms always give slightly different results in each run. Include statistical analysis aswell

The results seems to be manipulated, since there is no chance to have such an improvement in the quantitative and qualitative results using the proposed approach. If not, substantiate clearly.

6. PLOS authors have the option to publish the peer review history of their article (what does this mean?). If published, this will include your full peer review and any attached files.

Reviewer #1: No

Reviewer #2: No

---

## [Author Response · Author response to Decision Letter 0]

26 May 2022

List of Responses

Dear Editors and Reviewers，

On behalf of my co-authors, we thank you very much for giving us an opportunity to revise our manuscript, we appreciate editor and reviewers very much for their positive and constructive comments and suggestions on our manuscript entitled “A color image contrast enhancement method based on improved PSO”. (ID: PONE-D-21-36731).

Those comments are all valuable and very helpful for revising and improving our paper, and they are also having important guiding significance to us researches. We have studied those comments carefully and have made correction which are marked in red in the paper, hoping they will meet the standard. The main corrections and the responds to the reviewer’s comments are as flowing:

Responds to the reviewer’s comments:

Reviewer #1:

1. [Response to comment]:

The authors need to include the quality metrics in abstract to give a clear picture about quality of the proposed work.

[Responds]:

Considering the reviewer’s suggestion, we have written the quality metrics in abstract, clearly describing the quality of the proposed work.

2. [Response to comment]:

What is the novelty of the proposed PSO compared with existing PSO?

[Responds]:

In the manuscript, we have explained the proposed PSO in this part according to the reviewer’s suggestion. In order to ensure the diversity of particles, the PSO in this paper adds a local optimal value selected by the topology structure. And reduce the time complexity, a sparse penalty term is added to minimize the fitness value. The particles are penalized, and the effectiveness of the proposed PSO is verified by experiments.

3. [Response to comment]:

Need to separate introduction part and related work (literature survey part).

[Responds]: 

Considering the reviewer’s suggestion, we have separated the introduction from the related work and added the related work of the literature survey.

4. [Response to comment]:

The author is expected to include a separate section for to discuss the metrics, PSNR SSIM IFC VIF FSIM.

[Responds]: 

According to the reviewer’s suggestion, we have already introduced objective quality evaluation metrics in the experimental section.

5. [Response to comment]:

Measure the efficiency of the work using qualitative metrics such as Mean Opinion Score for reference kindly refer the following papers

[1] Swarm intelligent based contrast enhancement algorithm with improved visual perception for color images

[2] Particle swarm optimisation aided weighted averaging fusion strategy for CT and MRI medical images

[3] An optimal weighted averaging fusion strategy for remotely sensed images

[4] Optimal fusion aided face recognition from visible and thermal face images.

[Responds]: 

In the experimental part, by referring to several of the above literatures, we have added a qualitative indicator (Mean Opinion Score, MOS) to measure the efficiency of the work.

Reviewer #2: 

1. [Response to comment]:

The language and presentation need to proof-read and refined.

[Responds]:

We have revised the grammar carefully to make the paper smoother and more informative, which was done by us as well as the editing company.

2.[ Response to comment]:

There is no novelty in the transformation function, fitness function etc. It is the same as in Ref [1] The improved PSO algorithm is made use to optimize the selection of parameters k, a, b, c. Authors did not provide any clear mathematical, analytical or theoretical proof to substantiate how the improved version could perform much better as compared with conventional PSO.

[Responds]:

In the manuscript, we have analyzed the four parameters of the improved PSO optimization are better than the traditional PSO optimization.

3. [Response to comment]:

Comparisons with recent methods are missing. Include comparisons with few state-of-the art non-metaheuristic algorithms as well.

[Responds]:

Considering the reviewer’s suggestion, we have compared in this paper three recent image enhancement methods, but not based on meta-heuristics.

4. [Response to comment]:

Details about the dataset used is missing.

[Responds]:

In the manuscript, we have completed the details of the dataset.

5. [Response to comment]:

Since a deep layered structure was used in the improved variant of PSO proposed, quantify the computational complexity of the entire algorithm

[Responds]:

Considering the Reviewer’s suggestion, we have quantified the computational complexity of the proposed deep-structured particle swarm optimization.

6. [Response to comment]:

The deep layered structure of proposed algorithm is unclear. How are the parameters included in it tuned and selected? Are they image dependent?

[Responds]:

In the manuscript, we have detailed the deep topology, how the parameters are tuned and selected. They are not dependent on images.

7. [Response to comment]:

Since metaheuristic algorithms always give slightly different results in each run. Include statistical analysis as well. The results seem to be manipulated, since there is no chance to have such an improvement in the quantitative and qualitative results using the proposed approach. If not, substantiate clearly.

[Responds]:

Since metaheuristic algorithms always give slightly different results in each run. In this paper, the results are obtained after obtaining the average value through multiple experiments, and the experimental image is the optimal value obtained after optimization.

Responds to the Editor’s comments:

1. [Response to comment]:

You note that your data are available within the Supporting Information files, but no such files have been included with your submission. At this time, we ask that you please upload your minimal data set as a Supporting Information file, or to a public repository such as Figshare or Dryad.

Please also ensure that when you upload your file you include separate captions for your supplementary files at the end of your manuscript.

[Responds]:

We have uploaded the supporting information and added a separate title to the Supporting Information file at the end of the manuscript.

2. [Response to comment]:

We note that Figures 4 and 5 in your submission contain copyrighted images. All PLOS content is published under the Creative Commons Attribution License (CC BY 4.0), which means that the manuscript, images, and Supporting Information files will be freely available online, and any third party is permitted to access, download, copy, distribute, and use these materials in any way, even commercially, with proper attribution. 

a) You may seek permission from the original copyright holder of Figures 4 and 5 to publish the content specifically under the CC BY 4.0 license.

[Responds]:

In the manuscript, the copyrighted images contained in Figures 4 and 5 in the submission have been removed and replaced with images from our homemade dataset, under the CC BY 4.0 license.All images are completely taken by myself and do not involve copyright information.

Thank you for your constructive comments. 

We tried our best to improve the manuscript and made some changes in the manuscript. These changes will not influence the content and framework of the paper. We need to show our heartfelt appreciation for Editors/Reviewers’ efforts, and hope that the correction will meet with approval.

Once again, thank you very much for your comments and suggestions.

Yours sincerely,

Ren Yongfeng

E-mail: 503212590@qq.com

---

## [Decision Letter · Decision Letter 1]

22 Aug 2022

A color image contrast enhancement method based on improved PSO

PONE-D-21-36731R1

Dear Dr. Ren,

We’re pleased to inform you that your manuscript has been judged scientifically suitable for publication and will be formally accepted for publication once it meets all outstanding technical requirements.

Kind regards,

Diego Oliva

Academic Editor

PLOS ONE

Additional Editor Comments (optional):

Reviewers' comments:

Reviewer's Responses to Questions

**Comments to the Author**

1. If the authors have adequately addressed your comments raised in a previous round of review and you feel that this manuscript is now acceptable for publication, you may indicate that here to bypass the “Comments to the Author” section, enter your conflict of interest statement in the “Confidential to Editor” section, and submit your "Accept" recommendation.

Reviewer #1: All comments have been addressed

2. Is the manuscript technically sound, and do the data support the conclusions?

Reviewer #1: Yes

3. Has the statistical analysis been performed appropriately and rigorously? 

Reviewer #1: Yes

4. Have the authors made all data underlying the findings in their manuscript fully available?

Reviewer #1: Yes

5. Is the manuscript presented in an intelligible fashion and written in standard English?

Reviewer #1: Yes

6. Review Comments to the Author

Reviewer #1: (No Response)

7. PLOS authors have the option to publish the peer review history of their article (what does this mean?). If published, this will include your full peer review and any attached files.

Reviewer #1: **Yes: **Dr.K.Madheswari

---

## [Editor Report · Acceptance letter]

24 Aug 2022

PONE-D-21-36731R1 

A color image contrast enhancement method based on improved PSO 

Dear Dr. Ren:

I'm pleased to inform you that your manuscript has been deemed suitable for publication in PLOS ONE. Congratulations! Your manuscript is now with our production department. 

Kind regards, 

on behalf of

Dr. Diego Oliva 

Academic Editor

PLOS ONE